# Bone Marrow-Suppressive Treatment in Children Is Associated with Diminished IFN-γ Response from T Cells upon Polyclonal and Varicella Zoster Virus Peptide Stimulation

**DOI:** 10.3390/ijms25136960

**Published:** 2024-06-26

**Authors:** Eva Tiselius, Emil Sundberg, Hanna Andersson, Anna Höbinger, Peter Jahnmatz, Arja Harila, Josefine Palle, Anna Nilsson, Shanie Saghafian-Hedengren

**Affiliations:** 1Department of Women’s and Children’s Health, Division of Pediatric Oncology and Pediatric Surgery, Karolinska Institutet, 171 77 Stockholm, Sweden; eva.tiselius.2@ki.se (E.T.); shanie.hedengren@ki.se (S.S.-H.); 2Department of Women’s and Children’s Health, Uppsala University, 751 05 Uppsala, Sweden; emil.sundberg@uu.se (E.S.); arja.harila@uu.se (A.H.); josefine.palle@akademiska.se (J.P.); 3Mabtech AB, 131 52 Nacka Strand, Sweden; 4Department of Children’s Oncology and Hematology, Uppsala University Hospital, 751 85 Uppsala, Sweden

**Keywords:** memory T cell, varicella, IFN-γ, children, lymphoid malignancy

## Abstract

Severe haematological diseases and lymphoid malignancies require bone marrow (BM)-suppressive treatments. Knowledge regarding the impact of BM-suppressive treatments on children’s memory T cells is very limited. Memory T cells play a crucial role in defending against herpesviruses, which is particularly relevant in paediatric cancer care. We studied 53 children in total; 34 with cancer and 2 with severe haematological disorders, with some receiving BM-suppressive treatment with or without allogeneic–haematopoietic stem cell transplantation (allo-HSCT), alongside 17 healthy controls. We focused on peripheral blood proportions of memory T-cell subsets using flow cytometry and analysed cytokine-secreting T cells with a four-parameter FluoroSpot assay in response to T-cell mitogen and varicella zoster virus (VZV) peptides. Patients on BM-suppressive treatment showed increased clusters of differentiation (CD)4^+^ and CD8^+^ effector memory (TEM)/terminally differentiated effector (TEFF) T cells compared to the healthy controls. They also exhibited, amongst other things, when compared to the healthy controls, a reduced total number of cytokine-secreting cells, by means of interferon (IFN)-γ, interleukin (IL)-17A, IL-10, and IL-22, following mitogen activation. A diminished IFN-γ response among the children with BM-suppressive treatment was observed upon VZV-peptide stimulation, compared to the healthy children. Collectively, the findings herein indicate that the children who are undergoing or have finished BM-suppressive treatment display qualitative differences in their T-cell memory compartment, potentially increasing their susceptibility to severe viral infections and impacting their immunotherapy, which relies on the functional ability of autologous T cells.

## 1. Introduction

It is well established that T-cell memory is generated throughout childhood and that, upon birth, the T-cell compartment in peripheral blood consists primarily of naïve T cells. The T-cell compartment then gradually shifts to contain a larger proportion of memory T cells with age and exposure to antigens [1]. Memory T cells are crudely divided into central memory (TCM), effector memory (TEM), and terminally differentiated effector (TEFF) T cells. These subsets differ by means of proliferative and tissue homing ability. Along with antigen exposure, memory T cells can eventually differentiate to TEFFs, which is accompanied by a loss of proliferative potential and a gain of specific tissue-homing and functional abilities, as illustrated in Figure 1 [1,2,3]. It has been shown through comparisons of cell populations, as well as cytokine responses and expressions, that our immune system is largely shaped by non-hereditary factors [1,4,5], of which an important factor is herpesvirus infections [5,6,7,8,9]. Following a self-limiting primary infection in a healthy host, herpesviruses become latent and persist for life. Intermittent reactivation, which is often subclinical, ensures viral spread. Though latency in immunocompetent carriers is mostly asymptomatic, the immune system is highly active in controlling the latency, resulting in the emergence of a marked pool of memory CD4^+^ and CD8^+^ T cells in the host [10]. However, with immunosuppression, immune control can be perturbed and lead to reactivation of the herpesviruses, which can become life-threatening. For instance, among children with oncological or haematological disorders, varicella zoster virus (VZV), which causes chickenpox as the primary infection and herpes zoster (HZ) upon reactivation, can lead to treatment delay, secondary bacterial infections, or, in rare cases, severe life-threatening infections [11,12]. It has been shown that a third of children treated for acute lymphoblastic leukaemia (ALL) suffer from a reactivation of VZV [12]. A similar incidence rate of VZV reactivation has also been reported in children after haematopoietic stem cell transplantation (HSCT) [13]. Collectively, the above findings suggest that bone marrow (BM)-suppressive treatments may negatively impact VZV immunity.

Most of the studies regarding T-cell responses against herpesviruses in humans have been conducted in adults, whereas those on T-cell responses to VZV, in healthy and immunosuppressed children, are few or lacking. By measuring cell proliferation, Haining et al. reported preserved proliferative memory T-cell responses against VZV in children treated for ALL [14]. However, whether additional functional readouts, including cytokine release, were also preserved was not examined in that setting. To this end, we utilised a multiparameter FluoroSpot assay to quantify T cells releasing different cytokines, upon polyclonal or antigen-specific activation, in a cohort of paediatric haematology–oncology patients, among whom a subset was treated with BM-suppressive therapy.

## 2. Results

### 2.1. Memory T-Cell Subsets Are Enriched While Naïve T Cells Are Diminished in Children Exposed to Bone Marrow-Suppressive Treatment

The different T-cell populations were first examined through flow cytometry (Figure 2A) according to groups I-III (Figure 2B). To illustrate the impact of age, the proportions of T-cell subsets in each group were also plotted against age as a continuous variable (Figure 2B). No significant differences were found in the bulk CD4^+^ and CD8^+^ T-cell proportions between the groups. Both the CD4^+^ and CD8^+^ naïve T-cell proportions were significantly lower in group III compared to group I (70.5% (23.2–84.5%) vs. 21.2% (2.0–73.8%), *p* < 0.001, and 59.2% (38.4–90.6%) vs. 10.2% (0.5–86.9%), *p* < 0.01, respectively). The X/Y graphs illustrate how this effect may partly be explained by the increasing age in all groups. In this context, a more pronounced decrease in the proportion of naïve CD8^+^ T cells with increasing age in group III compared to the remaining groups is seen (Figure 2B). Furthermore, the proportions of TEM/TEFF cells among CD4^+^ T cells were significantly lower in group I compared to groups II and III (respectively, as follows: 5.5% (1.3–40.4%) vs. 24.4% (3.2–67.6%) for group II, *p* < 0.05, and vs. 47.3% (3.8–68.8%) for group III, *p* < 0.001). The CD8^+^ TEM/TEFF proportions were also significantly lower in group I compared to group III (7.6% (1.9–24.1%) vs. 45.4% (2.3–70.7%), *p* < 0.001). Once again, the X/Y plots reflect the impact of age, where higher proportions of TEM/TEFF populations are seen with increasing age. Nonetheless, the linear regression also suggests that group III had a more noticeable increase in CD8^+^ TEM/TEFF cells compared to group I. The proportions of CD4^+^ TCMs were significantly higher in group III compared to group I (18.0% (7.3–35.7%) vs. 8.6% (4.6–29.6%), *p* < 0.01). The influence of age on the TCM CD4^+^ proportions within the groups was not as clear as for the naïve and TEM/TEFF subsets (Figure 2B). There were no differences for the CD8^+^ TCM cell proportions among the groups, or in relation to age (Figure 2B). Finally, no differences were seen among the groups regarding the proportions of regulatory T cells (Appendix A). Two subjects within group III had received CAR-T-cell therapy, which did not allow for the stratification of the data for statistical analysis. Overall, the T-cell populations of these subjects did not deviate from the remaining individuals in group III.

### 2.2. Children with Bone Marrow-Suppressive Treatment Display Diminished Interferon-γ Producing Cells following Polyclonal and Varicella Zoster Virus-Specific T-Cell Activation

The functional status of the T cells was assessed through the simultaneous quantification of IL-22-, interferon (IFN)-γ-, IL-10-, and IL17-A-secreting cells upon T-cell mitogen (i.e., polyclonal activation) staphylococcal enterotoxin A (SEA), or VZV-specific (i.e., antigen-specific activation) stimulation (Figure 3A). First, spot-forming cell (SFC) counts for all cytokines were pooled for individuals with a complete assay that included all four cytokines. The number of total SFCs differed among the groups for both polyclonal and antigen-specific activation (Figure 3B). Group I had significantly more total SFCs upon SEA stimulation compared to group III (6245 (490–11,945) vs. 1080 (11–12,800), *p* < 0.05). Upon VZV stimulation, groups I and II had a similar total SFC count, whereas group III had a markedly lower response. This shift became significant when comparing the total SFCs following VZV stimuli in group II with those in group III (183 (30–3181) vs. 47 (9–368), *p* < 0.05, Figure 3B).

Next, the percentage of SFCs producing either type of cytokine from the donors with a complete cytokine dataset for IFN-γ, IL-17A, IL-22, and IL-10 was assessed upon activation of their peripheral blood mononuclear cells (PBMCs, Figure 3C). The overall cytokine responses to SEA (upper panel) were dominated by IFN-γ in all groups, with the remaining cytokines accounting for only 16–23% of secreting cells. However, a greater variation in the cytokine response was found among the groups following stimulation with VZV peptides (Figure 3C, lower panel). The proportions of IFN-γ-secreting cells for children in groups II and III were noticeably lower compared to group I. In group III, IFN-γ SFCs only accounted for 4% of secreting cells, whereas the majority (63%) consisted of IL-22 SFCs instead (Figure 3A). Significant differences after VZV peptide activation were found for the proportions of IL-22-secreting cells between groups I and III (*p* < 0.05), for IFN-γ-secreting cells between groups I and III (*p* < 0.01), and for IL-17A-secreting cells between groups I and III (*p* < 0.01) and between groups II and III (*p* < 0.05). The data for each specific cytokine SFC were also compared among the groups (Figure 3D). No significant differences were found for IL-22. For IFN-γ, the SFCs were significantly lower in group III compared to group I upon both SEA and VZV stimulation (1035 (0–12,380) vs. 5240 (375–9775), *p* < 0.05 and 2 (0–308) vs. 53 (5–351), *p* < 0.001, Figure 3D). In response to VZV, group III also had significantly lower ratios of IFN-γ compared to group II (2 (0–308) vs. 28 (2–602), *p* < 0.05, Figure 3D). The IFN-γ SFC results were further stratified for group III based on allo-HSCT, which revealed no clear differences (Appendix A). For IL-10, group III had a significantly lower ratio of IL-10 SFCs compared to group I upon SEA stimulation (80 (0–1570) vs. 543 (15–2730), *p* < 0.01), whereas no significant differences were seen following VZV stimuli (Figure 3D). When quantifying IL-17A SFC, no significant differences were found among the groups upon VZV stimuli (Figure 3D). However, upon SEA stimuli, group III had a significantly lower proportion of IL-17A SFCs compared to group I (15 (0–300) vs. 165 (10–515), *p* < 0.01).

Finally, the volume and intensity of each spot, quantified as the relative spot volume (RSV), were used as an additional quantitative marker of cytokine secretion (Figure 4A). Interestingly, we observed that the average RSV for IFN-γ in response to VZV was markedly lower in group III; specifically, between groups I and III (7810 (1407–29,184) vs. 462 (0–11,466), *p* < 0.0001, Figure 4A) as well as between groups II and III (6247 (247–32,453) vs. 462 (0–11,466), *p* < 0.05, Figure 4A). The IFN-γ RSV results were stratified for group III based on allo-HSCT, which revealed no clear differences between patients with or without allo-HSCT (Appendix A). Furthermore, the average RSV for IL-17A was significantly lower in both groups II and III compared to that in group I (2733 (0–1193) vs. 8998 (0–20,163), *p* < 0.05 and 934 (0–42,877) vs. 8998 (0–20,163), *p* < 0.01 respectively) upon polyclonal stimulation. No significant differences were found in RSV for IL-17A following VZV stimulation. The RSV data for IL-22 and IL-10 did not significantly differ among the study groups (Figure 4B).

## 3. Discussion

VZV infection poses challenges for paediatric oncologists, as primary infection can lead to critical illness and treatment delays. Furthermore, some immunocompromised children face higher VZV reactivation risks. We hypothesise that BM-suppressive treatment has a suppressive effect on memory T-cell function. Currently, it is difficult to predict the functional capabilities of T cells based solely on their immunophenotype. However, some cytokines can be associated with distinct subtypes of functional T cells [16]. Considering this, we opted to use the multiparameter FluoroSpot method, as it is highly sensitive and specific for the detection and quantification of immune cells that secrete cytokines associated with functionally distinct T-cell subsets [16]. Our previous findings highlighted IFN-γ SFC dominance in healthy children’s VZV-specific memory T-cell responses [15]. Here, we also explored polyclonal and VZV-specific responses in children exposed to BM-suppressive treatment. Prior studies have noted altered T-cell proportions in paediatric cancer, although to a lesser extent than for the B-cell populations (reviewed in [17,18]). In the current cohort, bulk CD4^+^ and CD8^+^ were similar among the groups. However, the proportions of naïve CD4^+^ and CD8^+^ cells were decreased in group III (children with BM-suppressive treatment with or without allo-HSCT), whilst accumulations of CD4^+^ and CD8^+^ TEM/TEFF cells, as well as the CD4^+^ TCM subset, were noted. Age relates to the accumulation of memory T cells [1,15], but it has also been shown that children on cancer treatment primarily exhibit a T-cell memory phenotype [14,19]. Interestingly, Das et al. showed that an accumulation of TEFF cells increases with each cycle of chemotherapy [20]. Therefore, it is possible that our results reflect variations in T-cell immunophenotype, explained both by age as well as diagnosis and its entailing treatment.

We further assessed the functional status of T cells and found that the frequency of cytokine-secreting cells differed significantly among the groups, as expected, due to the BM-suppressive therapy, where children in group III showed fewer responding cells. The cytokine profile upon polyclonal SEA activation relied heavily on IFN-γ, alluding to the expected Th1 response that is key to cellular immunity [21]. The relative amount of each T-cell cytokine to the total cytokine response following polyclonal T-cell activation (SEA) also aligned with those seen in a previous study in healthy children, where IFN-γ dominated [15]. In addition, it was observed that children in group III had significantly fewer IFN-γ-, IL-17A-, and IL-10-secreting cells following polyclonal activation compared to group I, accompanied by a diminished IL-17A secretion on a single-cell basis, measured as RSV. This decrease suggests a diminished T-cell response to polyclonal activation in children who have undergone the most intensive BM-suppressive treatments.

The VZV-specific T-cell memory cytokine profile showed a significantly lower proportion of IFN-γ in groups II and III, with a bias towards IL-22 and IL-17A secretion [22]. This shift in the cytokine profile is likely a result of a low number of IFN-γ-secreting cells, rather than an increase in the number of IL-22- or IL-17A-secreting cells. IFN-γ is the key cytokine secreted by Th1 cells, and a cytokine of significant importance during HZ [23]. Our data suggest that these cells are particularly affected in children belonging to group III, possibly due to the disease itself or to the allo-HSCT treatment affecting the lymphoid compartment. Another possible explanation could be differences in the drugs used in the applied chemotherapy protocols for different malignancies. Work by Mackall et al. in the 1990s demonstrated, for example, that the drug cyclophosphamide more effectively depleted lymphocytes compared to other drugs [24]. Children who had undergone allo-HSCT or CAR-T cells were also exposed to lymphodepleting chemotherapy, such as cyclophosphamide and fludarabine. There are also previous reports suggesting that 6-mercaptopurine, often used in the treatment of ALL, reduces the number of IFN-γ^+^ T cells in peripheral blood [25,26]. Conversely, Haining et al. showed that T-cell proliferation in response to VZV whole virus lysate and tetanus toxoid remained intact in children treated for ALL and was indeed greater than the proliferation in age-matched controls due to memory T-cell accumulation [14]. Herein, we demonstrate that children receiving BM-suppressive treatment, which includes lymphodepleting chemotherapy, may instead have a qualitative difference at the level of cytokine response, by means of diminished IFN-γ SFC, which could affect the immune response to VZV reactivation. In this context, it is also interesting that individuals with anti-IFN-γ autoantibodies often experience HZ [27].

We acknowledge that there are limitations of our study. Firstly, we studied a relatively small cohort, with diverse clinical characteristics, such as age, diagnosis and treatment regimens, and lines of therapy. For instance, group III contained children who received treatment for relapse. Furthermore, even if Hodgkin’s lymphoma is a haematological malignancy, in most cases, its treatment is stage-driven chemotherapy, with or without radiation [28]. In accordance, the treatment of one Hodgkin’s lymphoma subject in our study (found in group II) aligned more closely with solid tumour protocols due to a lack of relapse or refractoriness, which require high-dose chemotherapy and HSCT. On the other hand, the remaining subject with Hodgkin’s lymphoma was, at the time of sampling, undergoing pre-transplant salvage chemotherapy, which aligned with BM suppression (found in group III). A larger cohort would allow for an understanding of the impact of age, diagnosis, and treatment on memory T-cell responses. In addition, multiple sampling timepoints and longer follow-up would be central to enhancing our understanding of the dynamics of IFN-γ-producing T cells and their potential recovery as memory cells specific for VZV in children treated with BM-suppressive regimens.

In a broader context, the question demanding a more comprehensive answer pertains to how cancer and its different treatments impact the overall functional status of T-cell memory. While our understanding of the late effects of cancer treatment in children is gradually expanding, certain gaps still persist. Although we acknowledge that chemotherapy exerts a lasting effect on T cells [19], a thorough understanding of how chemotherapy, corticosteroids, allo-HSCT, and the underlying diagnoses collectively influence T-cell memory in the long term remains elusive [17]. Our findings may also have implications beyond infection immunity. Given that some novel therapies hinge on the functionality of autologous T cells, it is plausible that dysfunctional T cells [19], and/or an altered ratio between effector and regulatory T cells [29], could affect the efficacy of immunotherapeutic treatments. Understanding these dynamics becomes particularly relevant as we navigate the complexities of immunotherapeutic approaches and seek to enhance an anti-cancer response while reducing treatment toxicity, including infection sensitivity, in well-characterised study cohorts. Additional qualitative studies are imperative to unravelling the intricate interplay of disease, treatment, and their collective impact on T-cell function in the broader context of childhood health.

## 4. Materials and Methods

### 4.1. Cohort Description

This study was part of a prospective, single-centre longitudinal cohort study, with the primary aim of examining adaptive immune responses to COVID-19 in children between 0 and 18 years of age with cancer [30]. The current cross-sectional study addresses the secondary aim: to investigate memory responses to previously encountered pathogens in immunosuppressed children. Ethical approval was granted by the Swedish Ethical Review Authority (reference 2020-02154, approved 22 May 2020; amendment 2020-04672 approved 23 November 2020). Patients were randomly selected for the current study based on the availability of peripheral blood mononuclear cells (PBMCs) for the immunological assays and positive IgG status for VZV, which resulted in 39 children. Among these 39 cases, 3 children had a benign haematological disease, serving as controls, along with a separate cohort of 14 children without malignancies (previously described in [15]; granted ethical approval, Dnr 2020-05664 and Dnr 2014-1164-31/1), resulting in a total of 17 in the control group. Clinical characteristics, such as age, gender, diagnosis, immunosuppressive treatment, and previous allo-HSCT, were retrieved from electronic patient charts.

The patients were categorised based on their diagnosis and treatment (Table 1). This resulted in three groups: group I consisted of controls and children with anaemias having never been treated with chemotherapy (n = 17); group II consisted of patients with solid tumours, CNS tumours, and one subject diagnosed with Hodgkin’s lymphoma undergoing or having undergone immunosuppressive treatment (n = 16); and finally, group III consisted of 18 children with leukaemia (B-ALL n = 12, mixed-phenotype acute leukaemia n = 1, T-ALL n = 1) or lymphoblastic lymphomas (T-lymphoblastic lymphoma n = 1), 2 children with severe haematological disorders (aplastic anaemia n = 1, sickle cell anaemia n = 1), and 1 child with Hodgkin’s lymphoma, all of whom were undergoing or had undergone BM-suppressive treatment with or without allo-HSCT (group III, in total n = 20). Conditioning treatment pre-allo-HSCT was performed according to local guidelines; patients undergoing allo-HSCT for haematological malignancies received myeloablative conditioning, whereas conditioning for allo-HSCT for the treatment of benign haematological disease was decided by the treating physician to suit the patient and their diagnosis.

All demographic and clinical data are presented in Table 1. Although no significant differences were found among the groups, the children in group I had a lower median age compared to those in groups II and III (6 vs. 12 vs. 12 years old). Group I consisted of a larger proportion of females compared to groups II and III however, this was not statistically significant (70.5% females vs. 41.2% and 42.1% respectively). In group III, nine patients underwent allo-HSCT due to a severe haematological disease or malignancy, of which two subjects developed graft-versus-host disease (grade II). Among the subjects receiving allo-HSCT as part of their leukaemia treatment (n = 7), three received allo-HSCT in the first complete remission (CR1), two received allo-HSCT due to relapse, and two patients who received allo-HSCT were treated with chimeric antigen receptor (CAR) T-cell treatment following relapse (Table 1).

### 4.2. Handling of PBMCs and Cryopreservation

Blood samples were collected in CPT™ Mononuclear Cell Preparation Tubes with sodium heparin (BD Biosciences, Franklin Lakes, NJ, USA) during planned clinical sampling. The PBMCs and plasma were separated through centrifugation, after which the plasma was aliquoted and frozen at −80 °C. Next, the PBMCs were washed twice with phosphate-buffered saline (PBS, Gibco, Invitrogen, Carlsbad, CA, USA) before being counted with 0.4% trypan blue (Thermo Fisher Scientific, Waltham MA, USA). The PBMCs were then resuspended in a freezing medium, 10% dimethyl sulfoxide (DMSO, Invitrogen, Carlsbad, CA, USA) in foetal bovine serum (FBS, Sigma-Aldrich, St. Louis, MO, USA), at a concentration of 10^7^ viable cells/mL and frozen in liquid nitrogen. For the immunophenotyping of T-cell populations and FluoroSpot, the PBMCs were thawed and washed. The viable cell counts were once again determined with trypan blue staining and resuspended to a concentration of 2 × 10^6^ cells/mL in a cell culture medium (RPMI 1640, supplemented with 10% FCS, L-glutamine (2 mmol/L), penicillin G sodium (100 U/mL) and streptomycin sulphate (100 g/mL), all from Thermo Fisher Scientific). A portion of the PBMCs, 0.3–1.0 million, were assessed for flow cytometric analysis. The remaining were left to rest overnight at 37 °C and 5% CO_2,_ ahead of the FluoroSpot assay (described below). For the control cohort, the same procedure was carried out, apart from using Ficoll-Hypaque gradient centrifugation (GE Healthcare Bio-Sciences AB, Uppsala, Sweden) instead of CPT tubes.

### 4.3. Assessment of VZV IgG in Plasma through Enzyme-Linked Immunosorbent Assay (ELISA)

To determine the seropositivity for VZV, the Platelia VZV IgG kit assay (Bio-Rad Laboratories, Hercules, CA, USA) was carried out on all collected plasma, according to the manufacturer’s instructions. The optical densities were measured using the Multiskan FC microplate photometer (Thermo Scientific) at the appropriate wavelength.

### 4.4. Immunophenotyping of T-Cell Subsets by Flow Cytometry

The PBMCs underwent PBS washing and staining with a dead cell marker (DCM, Fixable Dead Cell Stain kit, Invitrogen) for 10 min at 4 °C in the dark. After washing, the cells were labelled with fluorescent-conjugated antibodies against CD3, CD4, CD8, CD25, CD57, CD95, CD127, CCR7, FOXP3, and a programmed cell death protein (PD)-1 (Biolegend, San Diego, CA, USA; Invitrogen, and BD Biosciences). For the memory T-cell panel, the samples were fixated with 4% paraformaldehyde for 5 min (Thermo Fischer Scientific), washed, and analysed. The control cohort received additional CD45RO staining. Regulatory T-cell immunostaining was exclusively performed on the oncological and haematological cohort. For regulatory T-cell staining, the PBMCs were fixated with the True-Nuclear™ Fix buffer (Biolegend) after surface staining with CD4, CD25, and CD127. True-Nuclear™ Perm Buffer (Biolegend) enabled permeabilisation for intracellular FOXP3 labelling, followed by final wash steps. Data were acquired with the Novocyte 3000 (ACEA Biosciences, Santa Clara, CA, USA) and analysed through the FlowJo software (v10.10.0, FlowJo LLC, Ashland, OR, USA). For statistical analysis, a threshold of 1000 live CD3^+^ single cells were established.

### 4.5. FluoroSpot for Simultaneous Detection of IL-22-, IFN-γ-, IL-10-, and IL-17A-Secreting Cells

The numbers of IL-22-, IFN-γ-, IL-10-, and IL17-A-secreting cells (from now on referred to as spot-forming cells, SFCs) were assessed with a human IL-22/IFN-γ/IL-10/IL1-7A FluoroSpot^PLUS^ kit (Mabtech AB, Nacka Strand, Sweden), according to the manufacturer’s instruction. Carried out in a sterile environment, pre-coated FluoroSpot plates were washed with PBS and blocked with a cell culture medium for one hour. Following the removal of the blocking medium, the T-cell mitogen Staphylococcal enterotoxin A (SEA, m *Staphylococcus aureus*, Sigma Aldrich, St. Louis, MO, USA), serving as polyclonal activator, was transferred to adequate wells at a final concentration of 20 ng/mL. Mixed VZV peptide pools, representing acute and latency MHC class I and II restricted epitopes, were used to assess specific T-cell responses to VZV. The VZV PepMix™ was aliquoted to a 1mg/mL final concentration for each of the IE63 and gE antigens per well (JPT Peptide Technologies GmbH, Berlin, Germany). Furthermore, a co-stimulatory anti-CD28 monoclonal antibody (Mabtech AB) was added, according to the manufacturer’s instructions, to all wells.

Following thawing and overnight rest, the PBMCs were washed with PBS, counted, and resuspended in a cell culture medium. A total of 80,000–100,000 live PBMCs were thereafter transferred to each well for T-cell stimulation with SEA. Then, 200,000–300,000 live PBMCs were transferred to appropriate wells, either for VZV-specific T-cell stimulation or wells containing a cell culture medium supplemented with the peptide pool diluent (DMSO, serving as a background control). Not all donors were assessed for all four cytokines due to the limited number of PBMCs after thawing. The plate was incubated for 48 h in 37 °C, humidified at 5% CO_2_ atmosphere. Following this, the plates were sequentially washed, and reagents for the detection of SFCs were added. Plates were read using the Mabtech IRIS™ ELISpot/FluoroSpot reader (Mabtech AB) equipped with wavelength-specific filters corresponding to DAPI, FITC, Cy3, and Cy5 spectrum for the detection of each cytokine separately. Each fluorescent spot represents the secretory footprint of a single cell. The frequency of spots was assessed with the Apex software version 1.1 (Mabtech AB). The data in the figures are presented as the SFCs per 10^6^ PBMCs, in mitogen or VZV antigen-stimulated wells, after the subtraction of SFCs in the background control wells. Furthermore, the relative spot volumes (RSVs) were also used to quantify the secretion using the Mabtech software Apex^TM^ (v2.0, Mabtech AB) by determining the volume and intensity of every SFC.

### 4.6. Statistics

For clinical characteristics, the Kruskal–Wallis test and Fisher’s exact test were used to compare age and gender among the groups, respectively. The Kruskal–Wallis test was also used for both flow cytometric and FluoroSpot data, together with a Dunn’s multiple comparisons test. The data are presented as the median (min–max), if not otherwise specified. X/Y tables for flow cytometric data, with a simple linear regression, were used solely as an illustrative method, with age as a continuous variable. For donors with a set of complete FluoroSpot data (i.e., for all four cytokines examined), the proportion of SFCs for each cytokine among the total SFCs per donor was calculated. The medians of these proportions within each group were then plotted into pie charts and statistically tested using the Kruskal–Wallis test with Dunn’s multiple comparisons test. A *p*-value of <0.05 was deemed statistically significant. All statistical analyses were carried out in Prism (v. 10.2.0, GraphPad Software, LLC, Boston, MA, USA).

## Figures and Tables

**Figure 1 ijms-25-06960-f001:**
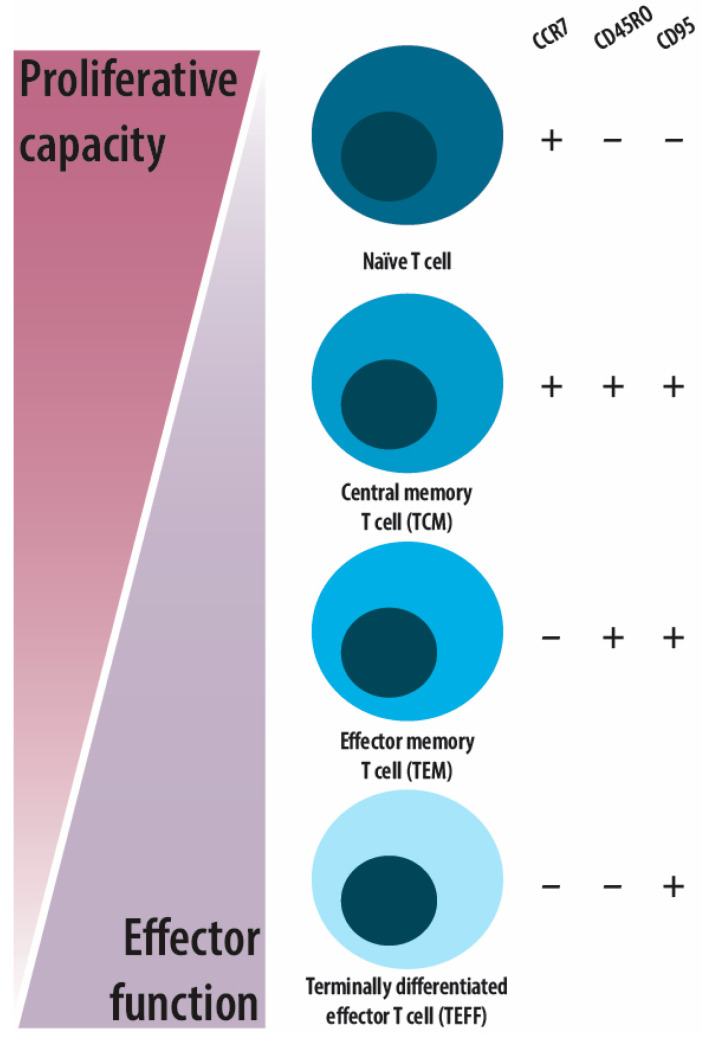
Simplified view of memory T cell subsets examined in this study, distinguished based on differential expression patterns of CCR7, CD45RO, and CD95. Naïve T cells are CCR7^+^CD45RO^−^CD95^−^; central memory T cells (TCM) are CCR7^+^CD45RO^+^CD95^+^; effector memory T cells (TEM) are CCR7^−^CD45RO^+^CD95^+^; and terminal effector T cells, also named terminally differentiated T cells (TEFF), are CCR7^−^CD45RO^−^CD95^+^, in accordance with previous findings by Gattinoni et al. [2]. As shown, the proliferative ability of T cells decreases along with differentiation and gain of effector function.

**Figure 2 ijms-25-06960-f002:**
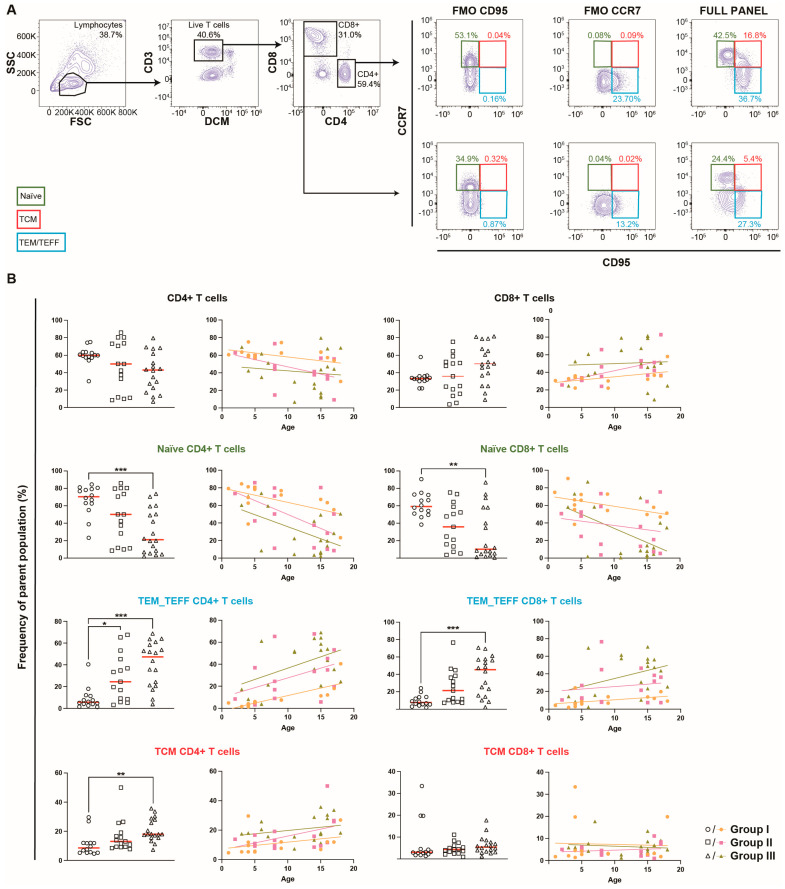
Variations in T-cell populations in peripheral venous blood among the groups. (**A**) Gating of naïve (CCR7^+^CD95^−^) and memory (CCR7^+/−^) T cells was performed on DCM^−^CD3^+^ single lymphocytes, which expressed either CD4 or CD8. Memory T cells were further sub-grouped into TCM (CCR7^+^CD95^+^) or TEM/TEFF (CCR7^−^CD95^+^) cells. Fluorescence minus one (FMO) controls were used for the gating of naïve and memory T-cell populations. Gating of the control cohort was similar, but included CD45RO, and is described in detail by Nilsson et al. [15]. (**B**) The differences in the proportions of T-cell subpopulations among the three groups (n = 47), and according to age, specifically looking at CD4^+^ and CD8^+^ bulk T cells, CD8^+^ and CD4^+^ naïve T-cell subsets, as well as memory T-cell subsets for both CD4^+^ and CD8^+^ T cells. A Kruskal–Wallis, with Dunn’s multiple comparison test, was used on the grouped scatter plots to determine any significant differences. Statistical significance was determined as *p* < 0.05 (* *p* < 0.05, ** *p* < 0.01, *** *p* < 0.001), and the median is shown as a red horizontal line.

**Figure 3 ijms-25-06960-f003:**
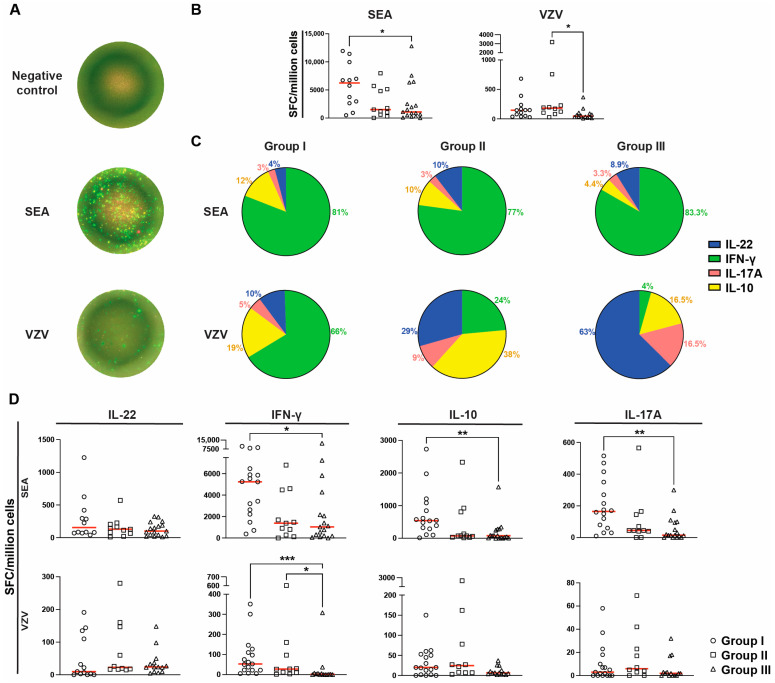
T-cell function in response to Staphylococcal enterotoxin A (SEA) and varicella zoster virus (VZV) peptides IE63 and gE in children with or without BM suppression. (**A**) Representative FluoroSpot wells at 1× magnification showing cellular expression of IL-22 (blue), IFN-γ (green), IL-10 (yellow), and IL-17A (red) when exposed to a negative control (dimethyl sulfoxide), T-cell mitogen SEA, or specific VZV peptides IE63 and gE. (**B**) Total counts per million of spot-forming cells (SFC) for individuals examined for all four cytokines following exposure to SEA (n = 40) and VZV (n = 38). (**C**) Cytokine profile depicting proportions of cytokine-producing T-cells based on the frequency—measured as SFCs/million cells—of IL-22-, IFN-γ-, IL-10-, and IL17-A-producing cells within each group. Significant differences after VZV exposure were found for the proportion of IL-22-secreting cells between groups I and III (*p* < 0.05), for IFN-γ-secreting cells between groups I and III (*p* < 0.01), and for IL-17A-secreting cells between groups I and III (*p* < 0.01) and between groups II and III (*p* < 0.05). (**D**) Frequencies of IL-22 (SEA n = 40, VZV n = 37), IFN-γ (SEA n = 44, VZV n = 41), IL-10 (SEA n = 44, VZV n = 41), and IL17-A (SEA n = 44, VZV n = 41) producing T cells shown at the polyclonal level, upon T-cell mitogen activation with SEA (top row) or at the antigen-specific level following activation with VZV peptides (bottom row). A Kruskal–Wallis test, with Dunn’s multiple comparison, was used to determine significant differences. Statistical significances were determined as *p* < 0.05 (* *p* < 0.05, ** *p* < 0.01, *** *p* < 0.001), and the median is shown as a red horizontal line.

**Figure 4 ijms-25-06960-f004:**
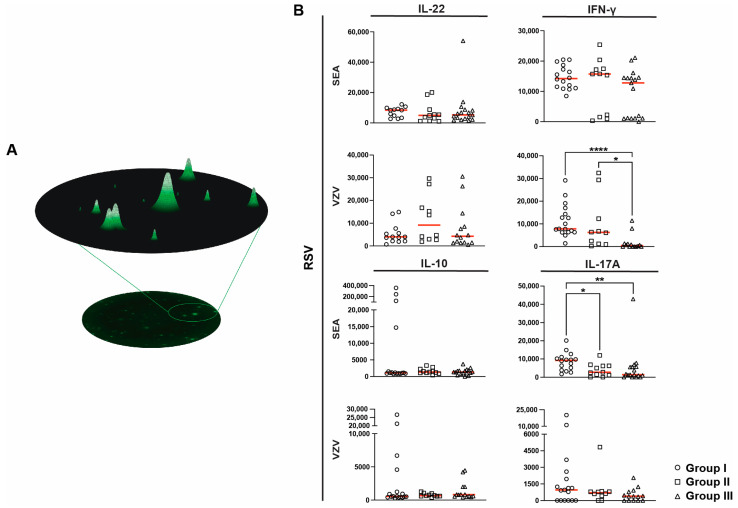
Residual spot volumes (RSV) upon stimulation with Staphylococcal enterotoxin A (SEA) or varicella zoster virus (VZV) peptides. (**A**) Schematic representation of RSV analysis. Using software algorithms, the RSV of individual spots was measured. These values were then determined as the area under the curve. (**B**) Differences in RSV from IL-22 (SEA n = 40, VZV n = 37), IFN-γ (SEA n = 44, VZV n = 40), IL-10 (SEA n = 44, VZV n = 40), and IL17-A (SEA n = 44, VZV n = 40) spot-forming cells following SEA or VZV stimuli among groups. Statistical analysis was performed using a Kruskal–Wallis test with Dunn’s multiple comparison. Statistical significance was determined as *p* < 0.05 (* *p* < 0.05, ** *p* < 0.01, **** *p* < 0.0001), and the median is shown as a red horizontal line.

**Table 1 ijms-25-06960-t001:** Clinical characteristics of the varicella zoster virus (VZV) seropositive children (n = 53) in the different subgroups (group I in green, group II in yellow and group III in orange).

Diagnosis	Sex (F/M)	Age (Years)	Ongoing Therapy (Months)	Follow-Up after Completing Therapy (Months)	Details
Control	F	17	N/A	N/A	
Control	F	1	N/A	N/A	
Control	F	3	N/A	N/A	
Control	F	4	N/A	N/A	
Control	F	4	N/A	N/A	
Control	F	4	N/A	N/A	
Control	F	4	N/A	N/A	
Control	F	10	N/A	N/A	
Control	F	10	N/A	N/A	
Control	F	15	N/A	N/A	
Control	F	16	N/A	N/A	
Control	M	5	N/A	N/A	
Control	M	5	N/A	N/A	
Control	M	6	N/A	N/A	
Anaemia, pyruvate kinase deficiency	M	18	N/A	N/A	
Thalassemia major	F	18	N/A	N/A	
Thalassemia major	M	9	N/A	N/A	
A-Rhabdomyosarcoma	M	14	>6	N/A	Relapse
Anaplastic ependymoma	M	5	Ongoing radiotherapy	N/A	
CNS germinoma	M	17	1	N/A	
Ewing sarcoma	M	14	At diagnosis	N/A	
Ewing sarcoma	M	16	>3	N/A	
Ganglioneuroma	M	5	N/A	>1	
Hodgkin’s lymphoma	F	16	>3	N/A	
Kaposiform lymphangiomatosis	M	17	>9	N/A	
Osteosarcoma	M	8	N/A	>1	
Osteosarcoma	M	15	1	N/A	
Pilocytic astrocytoma	F	5	N/A	>6	
Pilocytic astrocytoma	F	8	>6	N/A	
Retinoblastoma	F	2	N/A	>6	
Synovial sarcoma	F	8	>3	N/A	
Wilms’ tumour	F	12	1	N/A	
Wilms’ tumour	M	5	3	N/A	Relapse
B-ALL	F	2	>9	N/A	
B-ALL	F	4	>9	N/A	
B-ALL	F	6	N/A	>3	HSCT, CR1
B-ALL	F	7	>3	N/A	
B-ALL	F	11	>3	N/A	
B-ALL	M	6	>24	N/A	
B-ALL	M	11	N/A	>2	HSCT, CR1
B-ALL	M	12	N/A	>6	HSCT, relapse
B-ALL	M	15	N/A	>3	HSCT, relapse CAR-T
B-ALL	M	15	N/A	>3	HSCT, relapse, CAR-T
B-ALL	M	15	<1	N/A	
B-ALL	M	3	N/A	>12	HSCT, CR1
Chronic myeloid leukaemia	F	18	>6	N/A	
Aplastic anaemia	F	15	N/A	>12	HSCT
Sickle cell anaemia	M	14	N/A	>3	HSCT
Mixed-phenotype acute leukaemia	F	17	N/A	>3	
T-ALL	M	15	N/A	>1	HSCT, relapse
T-ALL	M	10	>12	N/A	
T-Lymphoblastic lymphoma	M	14	>3	N/A	
Hodgkin’s lymphoma	F	16	2	N/A	Relapse, pre-HSCT salvage chemo

Abbreviations: ALL, acute lymphoblastic leukaemia; HSCT, haematopoietic stem cell transplantation; CAR, chimeric antigen receptor; CNS, central nervous system, CR1, first complete remission.

## Data Availability

The data presented in this study are available on request from the corresponding author. The data are not publicly available due to patient confidentiality.

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
