# Peer review of "Bone Marrow-Suppressive Treatment in Children Is Associated with Diminished IFN-γ Response from T Cells upon Polyclonal and Varicella Zoster Virus Peptide Stimulation"

_ijms, 2024, doi:10.3390/ijms25136960_

Round 1

Reviewer 1 Report

Comments and Suggestions for Authors

This is a very important clinical study describing the dynamics of T-cell populations in children under treatment for cancer.  However, as an avid reader on such interesting subject I worry on the too concise nature of the text where my understanding was hampered by the lack of meaning for the many abbreviations used. This must be revised and corrected for the sake of clarity. Afterall most readers of IJMS are interested in molecular instead of cellular sciences.  In order to help readers and authors I prepared a list of suggestions for improvement.

1)ABSTRACT

Please, define HSCT, IFN, IL

2)INTRODUCTION

Provide an illustrative, colored figure with all subtypes possible for T-cell populations.

3)RESULTS

Line 74

Define PBMCs

Line 81

Define HSCT

Line 82

Define ALL

Line 105

Define TEM/TEFF

Line 125

Define TCM 

Line 127
Define CD27 and PD-1

Line 136

Do not ever use abbreviations in sub-titles! Please, provide SEA and VZV explicit meaning

Line 140

Define SFC

Thereafter provide an explanation on how subtypes are identified in peripheral venous blood

Line 176

Finally, SEA was defined! ALL ABBREVIATIONS NEED EXPLICIT DEFINITION FIRST TIME THEY APPEAR IN THE MANUSCRIPT

FIGURE 2

This poor-quality figure is not acceptable. Please, divide it in 3 separated figures and redraw each of them.

LIne 190

Define RSV

LIne 202

Finally, RSV was defined as residual spot volumes. Explain in METHODS the meaning of this determination.

LIne 244

Define HZ

Line 293

Finally, the meaning for PBMCs was given!

Appendix A

Provide definitions in loccus for SFC, SEA, VZV, etc IN THE FIGURE CAPTION.

Author Response

Please find our point-by-point response to both reviewers in the attachment.

Reviewer 2 Report

Comments and Suggestions for Authors

In my opinion, this topic is of interest to the community. 

However, several parts of this manuscript should be revised. There are mistakes  (or inconsistencies) in numerical data of the Materials section, that require correction.

Apart of that, the Statistical analysis seems to be reliable. In my opinion the Results and Discussion are also interesting.

My suggestions are as follows

The introduction satisfactory explains the context and aims of the work. Please do not mix introduction with the results: I suggest moving the last paragraph to the Results section.

Materials and Methods

Line 305 - please provide why two patients with Hodgkin's lymphoma were included in the group of solid tumours, not haematological malignancies

Line 307- please specify the type of haematological disorders

Line 308, 309- please be precise-what type of leukemia, B or T-cell Lymphoblastic lymphoma (this is important considering the topic of the work)

Line 308- In terms of patients after allo- HSCT: Conditioning regimens before transplantation were not specified. There is no information about Graft-versus-Host Disease after transplantation, which has a significant impact on the reconstitution of immunity         

Results:

Demographic and clinical cohort characteristics- this section as well as Table 1 should be part of Materials and Methods

 I don’t understand: In materials and methods as well as in table 1 we read as follows: the control group I has 17 patients, solid cancer group II- 17 patients and haematological group III- 19 patients. Meanwhile the data in the Results section seems to be incompatible with that; there are 14(?) children without malignancy –group I? (line-74) and 40 seropositive children, which somehow  gives 66.1% (what is the total count )

Line 82- ALL is also a severe haematological disease

The resolution of Figure 2 is not sufficient

Author Response

(The authors gave the same response as above.)
